# Cooling Performance Prediction of Particle-Based Radiative Cooling Film Considering Particle Size Distribution

**DOI:** 10.3390/mi15030292

**Published:** 2024-02-21

**Authors:** Jaehyun Lim, Junbo Jung, Jinsung Rho, Joong Bae Kim

**Affiliations:** 1Department of Mechanical Engineering, Kongju National University, Cheonan 31080, Republic of Korea; jhlim0212@smail.kongju.ac.kr (J.L.); junjoung3167@smail.kongju.ac.kr (J.J.); 2Department of Mechanical Engineering, Hanbat National University, Daejeon 34158, Republic of Korea; jinsung.rho@hanbat.ac.kr; 3Department of Mechanical and Automotive Engineering, Kongju National University, Cheonan 31080, Republic of Korea

**Keywords:** radiative cooling, micro-nanoparticle, regression analysis, image processing, chi-square test

## Abstract

Here, we present a novel protocol concept for quantifying the cooling performance of particle-based radiative cooling (PBRC). PBRC, known for its high flexibility and scalability, emerges as a promising method for practical applications. The cooling power, one of the cooling performance indexes, is the typical quantitative performance index, representing its cooling capability at the surface. One of the primary obstacles to predicting cooling power is the difficulty of simulating the non-uniform size and shape of micro-nanoparticles in the PBRC film. The present work aims to develop an accurate protocol for predicting the cooling power of PBRC film using image processing and regression analysis techniques. Specifically, the protocol considers the particle size distribution through circle object detection on SEM images and determines the probability density function based on a chi-square test. To validate the proposed protocol, a PBRC structure with PDMS/Al_2_O_3_ micro-nanoparticles is fabricated, and the proposed protocol precisely predicts the measured cooling power with a 7.8% error. Through this validation, the proposed protocol proves its potential and reliability for the design of PBRC.

## 1. Introduction

Various photonic structures for daytime radiative cooling, such as micro-grating and multi-layer structures, have achieved high cooling performance [1,2,3,4]. Still, there are substantial challenges in terms of flexibility, scalability, and cost. Recently, the particle-based radiative cooling (PBRC) structure method has been intensively developed as an alternative technique to photonic structure methods [5,6,7,8]. In general, the PBRC structure is composed of a polymer matrix, such as polydimethyl siloxane (PDMS); polymethyl methacrylate (PMMA); polymethyl pentene (TPX); and oxide micro-nanoparticles like ZrO_2_, Al_2_O_3_, and TiO_2_. Despite a low volume fraction of micro-nanoparticles, the composite can exhibit high reflectance, as well as high emissivity, through the refractive index mismatch between the matrix and oxide micro-nanoparticles [7,9,10,11,12,13]. Moreover, a metallic reflector is unnecessary in this structure when the film has enough thickness. Consequently, the PBRC method demonstrates strength in terms of scalability, flexibility, and cost-effectiveness [1].

In terms of the reflectance in the solar spectrum, the shape and size of oxide micro-nanoparticles, as well as their material compositions, are critical factors. For instance, Huang and Ruan [9] proposed a PBRC structure with acrylic resin and TiO_2_ micro-nanoparticles. In this study, the scattering efficiency, representing the reflectance of the particle, reached its maximum value at a wavelength of 0.5 μm when the radius of the particle is 0.1 μm. Moreover, it was observed that the peak of scattering efficiency shifted to a longer wavelength regime as the size of the particle increased. Thus, it is evident that the size of micro-nanoparticles is one of the crucial engineering factors influencing the optical properties of the mixture [14,15].

As described above, the size, shape, and material of micro-nanoparticles are key factors that can determine the performance of PBRC techniques. Hence, it is essential to design optimized topologies for micro-nanoparticles and fabricate uniform micro-nanoparticles for PBRC. However, several obstacles in the fabrication of ideal micro-nanoparticles are encountered during processes such as drying, washing, and heating treatment [16,17]. In general, metal oxide micro-nanoparticles are synthesized through chemical reduction reactions with precursors [18]. In this reaction, the concentration of the precursor and reaction time are critical factors determining the shapes of micro-nanoparticles, and the production of micro-nanoparticles with non-uniform shapes is inevitable due to practical problems. Even though the chemistry techniques for the ideal reaction still need to be studied, it is necessary to estimate the cooling performance of radiative cooling mixtures considering the non-uniform shape of micro-nanoparticles for now [19].

Regarding the non-uniform size and shape of micro-nanoparticles, various studies have been conducted to analyze their effects on radiative properties. Peoples et al. [20] investigated the size effect of TiO_2_ nanoparticles in a radiative cooling film. When employing a single-size model, they observed a maximum difference of 59% in total solar reflectance as the TiO_2_ nanoparticle diameter changed. Additionally, they found that the non-uniform size of TiO_2_ nanoparticles could result in approximately a 6.5% difference in total solar reflectance. Meanwhile, Cheng et al. [21] studied the non-uniform size effects of glass microspheres in the sky-window regime. In this study, the size of glass microspheres was measured using laser diffraction analysis. The findings revealed that the difference between uniform and non-uniform conditions led to a maximum 27.3% difference in spectral reflectance. As discussed in these studies, the size distribution of micro-nanoparticles has the potential to significantly impact the reflectance in the solar spectrum. Therefore, the size distribution of micro-nanoparticles should be taken into account when designing PBRC.

Typically, the size of micro-nanoparticles is determined using various methods, including the combined sieve–hydrometer method (SHM) [22], laser diffraction (LD) method [23], dynamic light-scattering (DLS) method, and microscopic method [24]. The SHM relies on the settling motion of particles, making it challenging to classify particles with diameters under 1 μm [22,25]. In contrast, indirect measuring techniques like LD and DLS can measure significantly smaller particles by using the light–matter interaction phenomenon. Although their measuring range spans from nanometers to millimeters, they have uncertainty issues induced by the unknown refractive index of particles [26,27]. The use of scanning electron microscopy (SEM) for particle shape–size analysis is a prominent method for direct observation of the sub-nanometer size scale [24,28]. Through SEM analysis, not only is it possible to image particles but also to measure their size. Nonetheless, this approach has its constraints, including a shortage of samples and the necessity of manual particle counting. Therefore, an accurate and simple prediction method based on improved SEM analysis is required for the design of PBRC.

In this study, we introduce a novel method for predicting the cooling performance of PBRC. To enhance accuracy and reduce processing time in micro-nanoparticle size measurement, we integrated a circle object detection technique with SEM analysis. In addition, we employed random number generation following a probability density function (PDF) for radiative property calculations using the Monte Carlo method with Mie-scattering theory. Furthermore, we selected the PDF carefully through a chi-square test, which assures the high similarity between the actual situation and the simulation. Finally, the radiative properties of the PBRC sample are computed, enabling an accurate prediction of its cooling performance. Comparing the predicted spectra to the measured spectra from a fabricated sample, it becomes evident that the proposed prediction method has precision and significant potential.

## 2. Methodology

### 2.1. Prediction Process for Cooling Performance

As mentioned before, this paper introduces a highly accurate and straightforward prediction method for the cooling performance of particle-based radiative cooling (PBRC). The overall process of the prediction method is illustrated in Figure 1. In Figure 1a, the prediction process begins with the imaging step of micro-nanoparticles using SEM and image processing. During this step, the morphology of micro-nanoparticles can be directly observed, and their size can be rapidly measured. It is important to note that image pre-processing for circle object detection and edge detection is required in this step. The probability density function (PDF) that can best reproduce the non-uniform distribution of micro-nanoparticles is chosen in the second step, as illustrated in Figure 1b. Herein, the PDF that best fits this situation is determined using the chi-square test statistic value. This is a novel step in our prediction method. This approach enables more accurate and reproducible simulations than those obtained using general PDFs in earlier studies.

Following the determination of the best-fit PDF, the spectral reflectance, absorptance, and transmittance of bulk mixtures are calculated using the Monte Carlo method as shown in Figure 1c. In each Mie-scattering theory calculation, the determined PDF is specifically used to generate a random particle size. In other words, the absorption and scattering efficiencies of micro-nanoparticles for the Monte Carlo method are provided by the calculation of Mie-scattering theory. In the end, the cooling performance of PBRC, which means net cooling power, is calculated based on the obtained spectral reflectance and absorptance. Figure 1d shows the mechanism of PBRC, as well as the equation for the net cooling power. Note that the net cooling power ($P_{net}$) is the sum of the thermal radiation to space ($P_{rad}$), the absorbed energy from the Sun ($P_{sun}$) and atmosphere ($P_{atm}$), and the transferred energy through conduction and convection ($P_{cc}$) [29].

### 2.2. Size Measurement of Micro-Nanoparticles

Herein, circle object detection is applied to SEM images to measure the size of micro-nanoparticles. Figure 2 shows detailed sub-processes of the image processing step, including circle object detection. While the use of SEM images on the fabricated sample is the most clear way to identify the morphology information of micro-nanoparticles, they have the potential to deform the shape of micro-nanoparticles during measurement due to the concentrated electron beam and high temperature. Hence, we prepared a grid specimen with only micro-nanoparticles for SEM analysis. In order to deposit micro-nanoparticles on the grid specimen, a dilute solution based on highly volatile ethanol and micro-nanoparticles was synthesized. In addition, the concentration of micro-nanoparticles was set to under 1% to avoid overlap of each particle during SEM analysis. In the final preparation step, a thin Pt layer was deposited onto the sample to enhance the quality of the micro-nanoparticle image.

In this paper, a PBRC film composed of PDMS and Al_2_O_3_ micro-nanoparticles is fabricated to validate the proposed prediction method. Figure 2a shows a SEM image of Al_2_O_3_ micro-nanoparticles employed in the radiative cooling film. As illustrated in Figure 2b,c, the edges of each micro-nanoparticle are detected by convolution operations with Sobel masks. The convolution operation with Sobel masks is one of the edge detection methods that utilizes the brightness change in two perpendicular directions [30]. For example, the grayscale values (0–255, black–white) from the SEM image are represented as Figure 2b, and the boundaries of the particle are revealed as white pixels through the convolution operation with the Sobel mask, as shown in Figure 2c.

Finally, the identified boundaries of micro-nanoparticles are determined through circle Hough transform (CHT). In general, there are several circle detection methods, including the perpendicular bisector method using the bisectors of chords [31], the least-squares method [32], the genetic algorithm [33], and CHT [34]. Among them, CHT is the most widely employed method in automatic particle size estimation. In principle, CHT generates numerous circles from the detected boundaries in convolution operations and detects the centers of circle objects based on their intersections. As illustrated in Figure 2d, the center and diameter of Al_2_O_3_ micro-nanoparticles are automatically identified through the CHT tool in the MATLAB Toolbox. A comparison of the original Al_2_O_3_ micro-nanoparticle SEM image and the circle object detection image is included in Appendix A.

### 2.3. Regression Analysis

In general, the lognormal distribution has been commonly used for statistical analysis of micro-nanoparticle sizes [35,36]. However, there is a potential for uncertainty in regression analysis due to a mismatch between the selected PDF and the raw data [37]. This paper newly introduces the chi-square test for the reliable selection of the PDF for the size distribution (population) in PBRC. When dealing with a large population, the chi-square test serves as a statistical hypothesis test capable of demonstrating the reliability between the PDF and the population. Chi-square ($\chi^{2}$) is defined as follows [38]:(1)$$\chi^{2}=\sum_{i=1}^{N}\frac{{\left(O_{i}-E_{i}\right)}^{2}}{E_{i}}$$
where $O_{i}$ and $E_{i}$ are the observed and expected frequency counts in each category, respectively, and *N* denotes the number of categories. Thus, it is clear that the chi-square value is smaller when the differences between the observed and expected values from the proposed PDF are decreased. In other words, it implies that the selection of the best-fit PDF by employing this criterion is possible. In this paper, we compute the chi-square value using the previously identified size distribution of micro-nanoparticles (2013 counts). We utilized representative PDFs from prior research [39,40,41,42,43,44,45,46,47,48], and the corresponding calculations are presented in Table 1.

As shown in Table 1, the results indicate that among the 11 types of PDFs, the Weibull distribution has the smallest chi-square value of 29.8, whereas the lognormal distribution has a value of 101.1. Consequently, we determined the Weibull distribution as the best fit and further confirmed its reliability by comparing randomly generated data, following the determined Weibull distribution, with the measured data.

Figure 3 presents the results of the regression analysis. The PDF of the Weibull distribution is represented in Figure 3, while x is the size of the particles. The Weibull distribution factors (a and b, calculated from measured data) are 1.63 and 2.45. The measured average size and standard deviation of Al_2_O_3_ particles are 1.447 μm and 0.631 μm, respectively. Upon generating 10^6^ counts of random data following the determined Weibull distribution, both the average size and the standard deviation show a 0.15% error compared to the measured data. Most importantly, this validation result consistently holds within repeated calculations. Consequently, we conclude that the PDF determined with the chi-square test is more reliable than others.

### 2.4. Prediction of Cooling Performance

As previously mentioned, the net cooling power of PBRC means its spontaneous cooling performance through heat exchanges, and it is calculated by summing the heat fluxes surrounding the mixture as follows: (2)$$P_{net}\left(T\right)=P_{rad}\left(T\right)-P_{atm}\left(T_{atm}\right)-P_{sun}-P_{cc}$$
where the net cooling power ($P_{net}$) is the sum of the thermal radiation to space ($P_{rad}$), the absorbed energy from the Sun ($P_{sun}$) and atmosphere ($P_{atm}$), and the transferred energy through conduction and convection ($P_{cc}$). Here, (*T*) and ($T_{atm}$) represent the surface temperature of the PBRC and the temperature of the atmosphere, respectively. The detailed calculation process is described in Appendix A. In short, it is obvious that the focus of this paper lies in prediction of PBRC considering the size distribution of micro-nanoparticles. Thus, this paper deals with the prediction of radiative properties—specifically, spectral reflectance and absorptance in the solar spectrum (0.3–2.5 μm) and the sky-window regime (8–13 μm).

The radiative properties, including reflectance and absorptance, are computed using the Monte-Carlo method. This method is commonly employed to calculate the radiative properties of heterogeneous materials such as PBRC film. The Monte Carlo method statistically accounts for the independent interaction between matter and photons when calculating radiative properties [49]. More specific information about the Monte Carlo method and its critical factors is provided in Appendix A. Theoretically, its principle relies on repeated random generation so that it is possible to embed the determined PDF within this random sample generation approach. Specifically, micro-nanoparticles with diameters following the determined PDF are randomly generated during each independent interaction. In each independent interaction, the optical properties of single particles, such as absorption and scattering efficiencies, are calculated using the Mie-scattering theory. As a result, the radiative properties of PBRC film are derived from these calculations. Eventually, the cooling performance of PBRC film is predicted by considering the particle size distribution following the determined PDF.

## 3. Results

### 3.1. Effects of Micro-Nanoparticles on Radiative Properties

First, we analyze the effects of micro-nanoparticles on the radiative properties of particle-based radiative cooling (PBRC) film. As mentioned earlier, the PBRC film was designed to contain the mixture of Al_2_O_3_ micro-nanoparticles and PDMS to validate the proposed method. Similar to typical PBRC film, the mixture of PDMS and Al_2_O_3_ micro-nanoparticles is placed on the aluminum (Al) substrate. The volume fraction of Al_2_O_3_ micro-nanoparticles is 20%, and the thickness of the film is 1 mm [50]. The diameter of Al_2_O_3_ micro-nanoparticles is constant as a measured average size (D_*avg*_ = 1.447 μm). Figure 4a shows the spectral reflectance of PBRC film, as well as the Al substrate in the solar spectrum (0.3–2.5 μm). The spectral reflectance of the case without Al_2_O_3_ micro-nanoparticles (bare PDMS only) is also shown in Figure 4a.

Even though the spectral reflectance of each case varies depending on the wavelength, it is possible to observe the effect of Al_2_O_3_ micro-nanoparticles in Figure 4a. In the visible spectrum (0.4–0.7 μm), the PDMS/Al_2_O_3_ case shows the highest reflectance among them, while the bare PDMS case shows the lowest reflectance, at approximately 0.81. The slightly lower spectral reflectance of bare PDMS compared to the Al substrate suggests that PDMS absorbs light in this range. However, its spectral reflectance in the visible spectrum is enhanced by mixing the Al_2_O_3_ micro-nanoparticles, as depicted in Figure 4a. This highlights the importance of particle selection. Unfortunately, significantly different tendencies in spectral reflectance are observed in the near-infrared spectrum. Here, the Al substrate shows the highest spectral reflectance compared to the others. This is due to PDMS possessing intrinsic absorption properties in the near-infrared spectrum, as well as the exponential decrease in the scattering efficiency of Al_2_O_3_ micro-nanoparticles, as shown in Figure 4b. This is reasonable, since bare PDMS and PDMS/Al_2_O_3_ cases exhibit absorption peaks at the same wavelengths, although differences in spectral reflectance are observed between them.

Figure 5a presents the spectral emissivity of each case in the sky-window regime. As explained earlier, thermal radiation through the sky-window regime is a crucial factor in achieving ideal radiative cooling, since this regime is transparent to space. Therefore, to achieve not only higher spectral reflectance in the solar spectrum but also higher spectral emissivity in the sky window is important. In Figure 5a, it is found that the PDMS/Al_2_O_3_ and bare PDMS cases show spectral emissivities over 0.81 in this regime. At the same time, the Al substrate case shows extremely low emissivity below 0.06. That is, the spectral emissivity increases as bare PDMS film is placed on the Al substrate. In addition, the slight difference between PDMS/Al_2_O_3_ and bare PDMS cases is related to radiative properties of Al_2_O_3_ micro-nanoparticles. The calculated scattering and absorption efficiency of single Al_2_O_3_ micro-nanoparticles by Mie-scattering theory is shown in Figure 5b. Even though the scattering efficiency is nearly zero until 10 μm, it is found that it increases gradually after that and has a peak at 12.3 μm. Since the scattering efficiency of micro-nanoparticles has negative effects on emissivity, a slight decrease in spectral emissivity is observed in this range [21,51].

Consequently, it is possible to validate the effects of micro-nanoparticles in both the solar spectrum and sky-window regime by using the prediction method proposed in this paper. In addition, micro-nanoparticles and medium effects on radiative properties can be analyzed by the scattering and absorption efficiency calculated using Mie-scattering theory.

### 3.2. Effects of Particle Size Distribution on Radiative Properties

By employing circle object detection and regression analysis, it has been established that the probability density function (PDF) follows a Weibull distribution. Using this PDF, it is possible to generate an imaginary statistical data set on the size of micro-nanoparticles. Here, the generated imaginary probability distribution function can accurately simulate the measured particle size distribution. Table 2 represents the particle size information corresponding to different cases. As shown in Table 2, several cases are assumed, like size following the Weibull distribution (D_*dis*,*Weibull*_), constant size with average (D_*avg*_), maximum size (D_*max*_), and minimum size (D_*min*_).

Figure 6a shows the predicted spectral reflectances in the solar spectrum in each case. While the detail of spectral reflectance varies depending on the wavelength regime, a consistent pattern of spectral reflectance is observed in each case. The averaged spectral reflectance of the D_*avg*_ in the visible regime is highest at 0.92, and it decreases in the order of D_*dis*_, D_*min*_, and D_*max*_. In addition, the disturbance of spectral reflectance is unusually observed in the D_*dis*_ case. To analyze these results effectively, it is essential to consider the scattering efficiency and the ratio of scattering to absorption efficiencies for individual micro-nanoparticles. This is particularly important since all cases were calculated under the same thickness and volume fraction conditions. Therefore, the optical properties of individual micro-nanoparticles were determined using the Mie-scattering theory, as depicted in Figure 6b.

First of all, it is found that the average-sized Al_2_O_3_ particle shows the highest scattering efficiency while keeping the absorption efficiency below 0.6. Additionally, a gradual decrease in efficiency is observed in the mid-infrared regime, with the peak of scattering efficiency found at 0.82 μm. Compared to the D_*avg*_ case, the D_*max*_ case shows lower scattering efficiency and higher absorption efficiency. According to these observations, we find that the ratio of scattering to absorption efficiencies in the D_*max*_ case is lower than in the D_*avg*_ case. For this reason, it is concluded that the enhancement of spectral reflectance in the D_*max*_ case is not as effective due to the increased absorption by the larger particles.

In the D_*mim*_ case, the enhancement of spectral reflectance is challenging to observe when compared to the bare PDMS case, as depicted in Figure 5a and Figure 6a. Furthermore, a slight decrease is noted below 0.4 μm in the minimum size case. Theoretically, this can be analyzed through Rayleigh scattering theory. In the minimum size case, the diameter of Al_2_O_3_ particles is 6 nm, representing an extremely small scale comparable to the wavelength of the solar spectrum. Consequently, this condition aligns with the Rayleigh scattering regime, where scattering phenomena rarely happen [52]. Appendix A has more details on small particle scattering and Rayleigh scattering phenomena. This is why the enhancement of spectral reflectance is quite low, as it is also possible to see in Figure 6b. In the D_*dis*_ case, a spectral reflectance similar to the average case is predicted, as shown in Figure 6a, since it contains various sizes of particles, including maximum-size particles. A disturbance pattern in spectral reflectance is observed, but it is evidently caused by the size distribution of Al_2_O_3_ micro-nanoparticles.

For the sky-window regime, Figure 7a illustrates the spectral emissivities for different cases, while Figure 7b presents the computed absorption and scattering efficiencies of individual micro-nanoparticles using the Mie-scattering theory. In contrast to the solar spectrum, there are small differences in spectral emissivities among the cases, and they are similar to the spectral emissivity of the bare PDMS case depicted in Figure 5a. The most substantial differences among them occur at 10 to 12 μm, where the D_*max*_ case shows the lowest spectral emissivity. In addition, the computed scattering and absorption efficiencies of the individual particles can suggest why the difference is occurring. Notably, the D_*max*_ case shows larger scattering and absorption efficiencies than other cases, with a significant increase from 9 μm as depicted in Figure 7b. Furthermore, the ratio of scattering to absorption efficiencies in the D_*max*_ case exceeds 1, despite other cases being below 1. This indicates that the scattering phenomenon predominates over absorption in this case. Consequently, it is inferred that the strong scattering effect contributes to the reduction in spectral emissivity in the D_*max*_ case.

### 3.3. Validation of the Prediction Method on Radiative Properties of Pbrc

To validate the proposed prediction method, a PBRC film is fabricated with PDMS/Al_2_O_3_ micro-nanoparticles, as shown in Figure 8. The detailed fabrication process is described in Appendix A. The Al_2_O_3_ micro-nanoparticles utilized in this film are consistent with those employed for circle object detection, and we made an effort to fabricate the PBRC film using the same volume fraction and thickness condition. The radiative properties of the fabricated sample were measured with ultraviolet-visible (UV-Vis) spectroscopy (Shimazu, Kyoto, Japan, UV-3600i plus) and Fourier transform infrared (FT-IR) spectroscopy (ABB Bomem, Quebec, QC, Canada, FTLA 2000 series). Figure 9 presents the spectral reflectance in the solar spectrum and the spectral emissivity in the sky-window regime with the predicted results.

As described in Figure 9, the predicted spectral reflectances and emissivities were computed by employing Weibull and exponential distribution, showing minimum and maximum chi-square values, respectively. In terms of averaged radiative properties in the solar spectrum and the sky-window regime, the measurement result shows 0.774 of the spectral reflectance and 0.943 of the spectral emissivity. When Weibull distribution is employed, 0.819 of the averaged reflectance in the solar spectrum and 0.94 of the averaged emissivity in the sky-window regime are predicted. Employing the exponential distribution, 0.779 of the averaged reflectance and 0.942 of the averaged emissivity are also predicted. Performing this comparison, it becomes evident that the prediction method relying on the chi-square test proves to be both reliable and valid. Nevertheless, remarkable discrepancies at several ranges are found between the measurement and prediction. This is because slight discrepancies in the volume fraction, thickness, and uniformity of the micro-nanoparticles are inevitable, despite efforts to fabricate the sample under ideal conditions. These discrepancies may also be the result of insufficient micro-nanoparticle images for determination of the PDF.

### 3.4. Cooling Performance Prediction Considering Particle Size Distribution

In the final step, we calculate the cooling performance of PBRC based on the predicted radiative properties, taking into account the particle size distribution. Here, cooling performance refers to the ability to cool down an object solely through surface attachment. Appendix A offers a detailed explanation of the cooling power calculation process using the predicted spectral reflectance and emissivity. Theoretically, positive cooling power can lead to the achievement of temperatures lower than the ambient temperature, while negative cooling power results in the surface temperature exceeding the ambient level. Note that this is a straightforward yet significant concept for understanding the prediction results.

Figure 10 presents the predicted cooling powers for different cases, characterized by specific particle size cases as explained before. For example, the D_*dis*,*Weibull*_ case determines the size of Al_2_O_3_ micro-nanoparticles using the Weibull distribution. In the D_*avg*_ case, the size is set as constant at the averaged diameter. The black line represents the calculated cooling power of the fabricated sample (38.3 W/m^2^).

Depending on the particle size case, the cooling power is calculated as follows: D_*avg*_ as 58.1 W/m^2^, D_*max*_ as −12.7 W/m^2^, D_*min*_ as −17.7 W/m^2^, D_*dis*,*Exp*_ as 25.6 W/m^2^, and D_*dis*,*Weibull*_ as 41.3 W/m^2^. This implies that cases of D_*max*_ and D_*mim*_, exhibiting negative cooling power, cannot possibly implement daytime passive radiative cooling. On the other hand, the D_*dis*,*Weibull*_, D_*dis*,*Exp*_, and D_*avg*_ cases exhibit positive cooling power in their predictions, indicating the potential to achieve temperatures lower than the ambient level. This means that the determination of cooling power dominantly relies on particle size conditions. Consequently, the consideration of particle size distribution is the most critical factor in PBRC design.

As shown in Figure 10, the D_*dis*,*Weibull*_ case predicts a cooling power of 41.3 W/m^2^ under the given conditions. In this case, the deviation between the measurement and prediction is 3 W/m^2^, with an error rate of 7.8%. Compared to other cases, it is the most precise prediction of the actual cooling power calculated from the measured spectra. However, in the D_*avg*_ and D_*dis*,*Exp*_ cases, positive cooling powers of 58.1 W/m^2^ and 25.6 W/m^2^ are predicted, respectively. Furthermore, the error rates in each case are 51.6% and 33.1%, indicating larger discrepancies in cooling power predictions compared to our proposed method. Employing our proposed method, the accuracy of theoretical cooing performance prediction of a state-of-the-art PBRC structure can be improved. In conclusion, the proposed method has the potential to predict the cooling performance of PBRC before fabrication, and it is a necessary technique for the design of an effective PBRC.

## 4. Conclusions

In this study, we propose a novel prediction method for the cooling performance of particle-based radiative cooling (PBRC), considering the size distribution of micro-nanoparticles. The overall process of the prediction method consists of scanning electron microscopy (SEM) image capture, image processing for circle object detection, a chi-square test, the Monte Carlo method with Mie-scattering theory, and calculation of cooling power. This method offers direct observation of micro-nanoparticle morphologies through SEM images and provides statistical data on micro-nanoparticle size through automated circle object detection. Ensuring high reliability and precision, the most suitable probability density function (PDF) is determined using quantitative criteria such as chi-square. Subsequently, the determined PDF with the best fit is integrated into the Monte Carlo method to estimate the radiative properties, like the spectral reflectance and the spectral emissivity. Finally, the cooling power of PBRC is calculated by substituting the estimated radiative properties. To experimentally validate the proposed method, we fabricated a PBRC structure with PDMS/Al_2_O_3_ micro-nanoparticles. The radiative properties of the fabricated sample were measured by a commercial measurement system. The cooling power was computed from those data. In comparing predicted and measured values, a remarkably accurate prediction with a 7.8% error was obtained, which is closer than other predictions. Consequently, it is proven that the proposed method has great potential for the design of effective PBRC and can also reduce trial and error in research.

## Figures and Tables

**Figure 1 micromachines-15-00292-f001:**
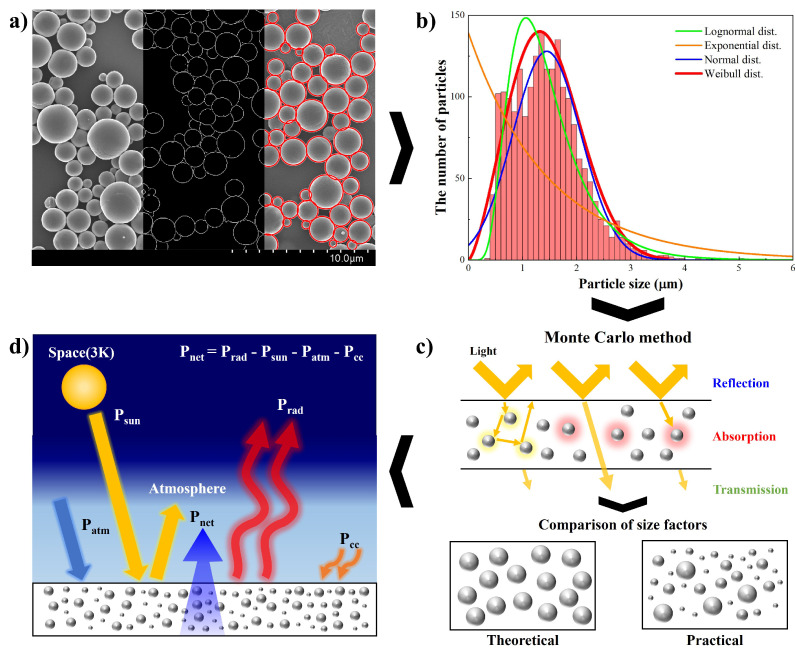
The overall process of the particle-based radiative cooling (PBRC) prediction method. (**a**) Size measurement process of micro-nanoparticles using SEM imaging and image processing. (**b**) Comparison of various probability density functions (PDFs). (**c**) Radiative property calculation using the Monte Carlo method. (**d**) Mechanism of PBRC cooling performance.

**Figure 2 micromachines-15-00292-f002:**
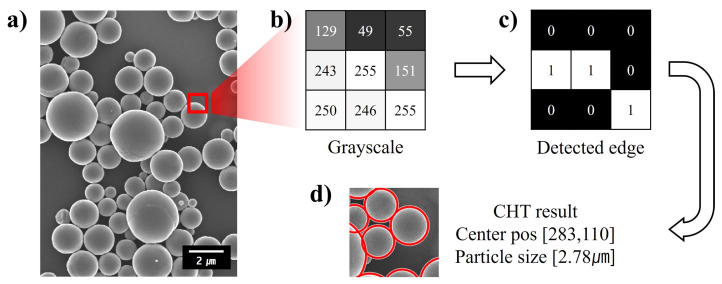
Size measurement process of Al_2_O_3_ micro-nanoparticles: (**a**) SEM image; (**b**) grayscale value of the SEM image; (**c**) edge detection through Sobel mask convolution; (**d**) result of circle object detection on the SEM image.

**Figure 3 micromachines-15-00292-f003:**
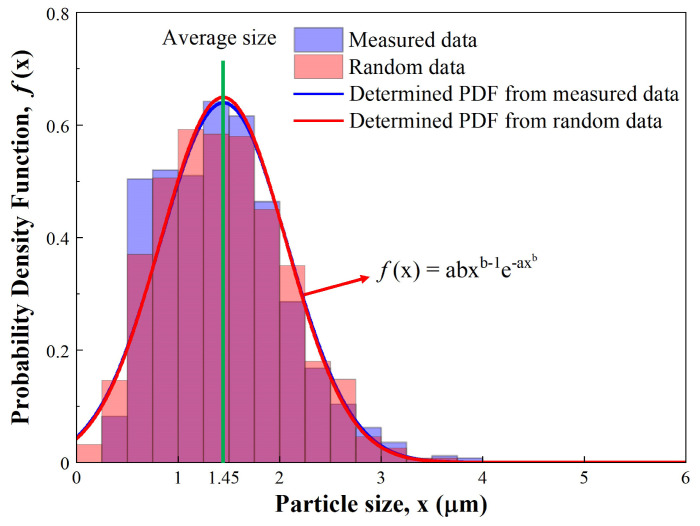
Weibull distribution probability density functions (PDFs) of measured and random data.

**Figure 4 micromachines-15-00292-f004:**
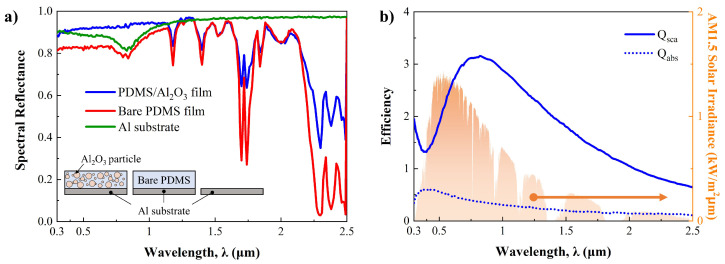
(**a**) Spectral reflectance of PBRC film, bare PDMS film (without Al_2_O_3_ micro-nanoparticles) and Al substrate in the solar spectrum. (**b**) Scattering (Q*_sca_*) and absorption (Q*_abs_*) efficiency of Al_2_O_3_ particle at averaged size (D*_avg_* = 1.447 μm) in the solar spectrum and AM1.5D solar irradiance.

**Figure 5 micromachines-15-00292-f005:**
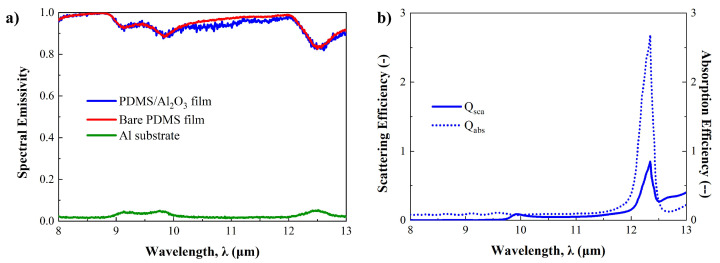
(**a**) Spectral emissivity of PBRC film, bare PDMS film (without Al_2_O_3_ micro-nanoparticles) and Al substrate in the sky-window regime. (**b**) Scattering and absorption efficiency of Al_2_O_3_ particle at averaged size (D*_avg_* = 1.447 μm) in the sky-window regime.

**Figure 6 micromachines-15-00292-f006:**
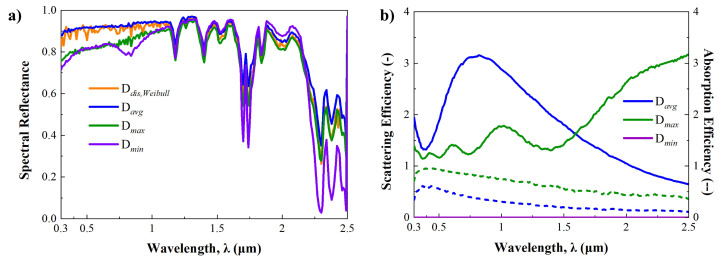
(**a**) Predicted spectral reflectance of each particle size case in the solar spectrum. (**b**) Scattering (solid line) and absorption (dotted line) efficiency according to each particle size case in the solar spectrum.

**Figure 7 micromachines-15-00292-f007:**
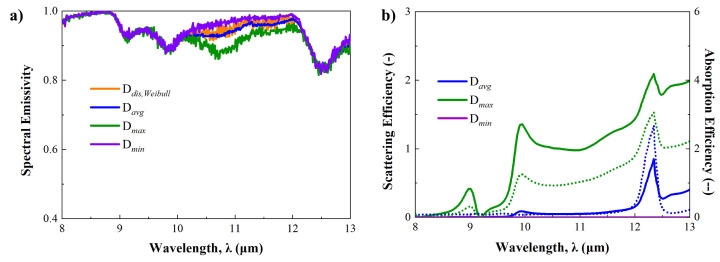
(**a**) Predicted spectral emissivity of each particle size case in the sky-window regime. (**b**) Scattering (solid line) and absorption (dotted line) efficiency according to each particle size case in the sky-window regime.

**Figure 8 micromachines-15-00292-f008:**
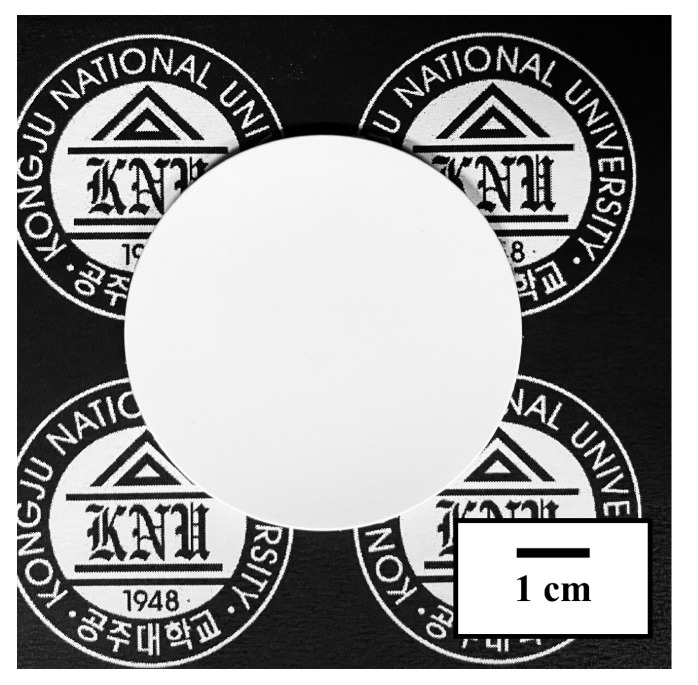
PDMS/Al_2_O_3_ sample film.

**Figure 9 micromachines-15-00292-f009:**
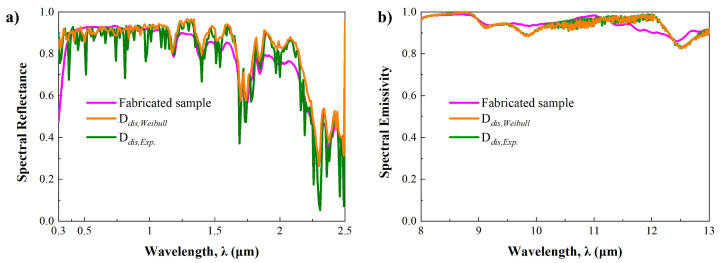
Radiative properties of fabricated PDMS/Al_2_O_3_ sample and Weibull (D_*dis*,*Weibull*_) and Exponential (D_*dis*,*Exp*_) distribution cases. (**a**) Spectral reflectance in the solar spectrum (**b**) Spectral emissivity in the sky-window regime.

**Figure 10 micromachines-15-00292-f010:**
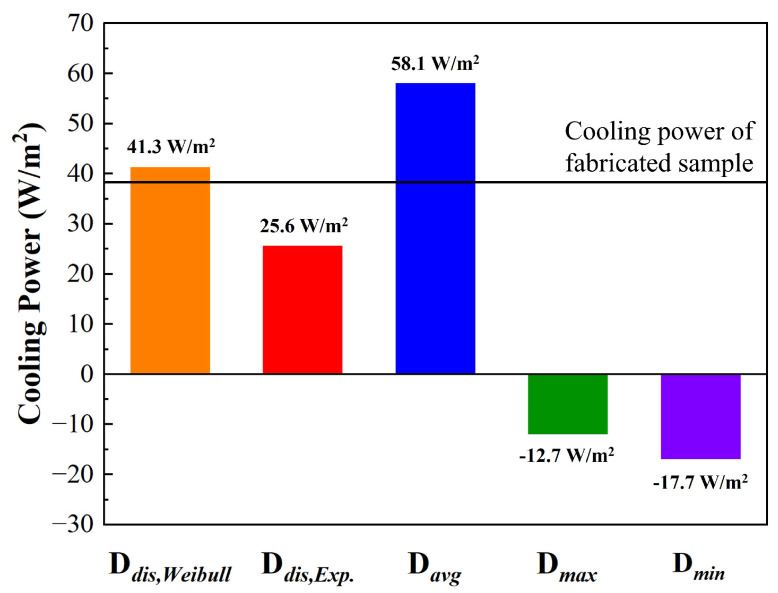
Predicted cooling powers of 5 particle size cases compared with the fabricated PBRC sample.

**Table 1 micromachines-15-00292-t001:** Result of chi-square test.

Distribution Type	$\chi^{2}$
Birnbaum–Saunders	97.2
Exponential	1362.2
Extreme Value	333.7
Generalized Pareto	850.5
Half Normal	691.7
Inverse Gaussian	105.6
LogLogistic	129.2
Lognormal	101.1
Normal	84.3
Rayleigh	92.3
Weibull	29.8

**Table 2 micromachines-15-00292-t002:** Information for each particle size case.

Case	Particle Size
D_*dis*,*Weibull*_	*f*(*x*) = $1.63\cdot 2.45x^{2.45-1}e^{-1.63x^{2.45}}$
D_*dis*,*Exp*._	*f*(*x*) = 1.44$e^{-1.44x}$
D*_avg_*	1.447 μm
D*_max_*	4.856 μm
D*_min_*	0.006 μm

## Data Availability

Data are contained within the article and Appendix A.

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
