# Peer review of "Cooling Performance Prediction of Particle-Based Radiative Cooling Film Considering Particle Size Distribution"

_micromachines, 2024, doi:10.3390/mi15030292_

Round 1

Reviewer 1 Report

Comments and Suggestions for Authors

There are some points in the manuscript that need to be addressed before acceptance. Here are comments;

1.     Line 40-41: “Poly….” should be written as “poly…”.

2.     Line 46: “Acrylic resin” should be written as “acrylic resin”.

3. The introduction part is quite lengthy with 8 paragraph. Considering condense it for better focus.

4.  In section 2, materials/chemical specifications of PDMS and Al2O3 for the experiment are missing. And how the author prepare PDMS/ Al2O3 sample? Please provide details.

5.     Please add a scale bar in Figure 2a.

6.    Figure 5a, the spectral reflectance of bare Al2O3 particles should be added for comparison.

7.  In the case of fabricate sample, what is the average size of Al2O3?

Author Response

Thank you for your careful comments again. We addressed all of the comments in our manuscript and described the details in the Author's response. Please refer to the attached file below.

Reviewer 2 Report

Comments and Suggestions for Authors

The article is technically robust and was a pleasure to read. I have a few minor suggestions to further enhance the manuscript.

Adhesion of Particles to Surfaces: Could you provide more details on the mechanism of particle adhesion to the surface? It would be beneficial to understand the factors influencing this interaction. Please consider citing relevant literature to support this statement, if available.
SEM Imaging Preparation: Was gold coating applied to the samples before SEM imaging? Clarification of sample preparation methods would enhance the reproducibility of the results.

Figure 2 Clarifications: The representation of only nine gray values in Figure 2 raises questions about the selection criteria. Were these values derived from a specific selection of pixels, or do they serve a symbolic purpose? Furthermore, how does one accurately identify the presence of particles atop other particles, considering potential limitations in edge detection by the algorithm?

Spectral Emissivity and Scattering Efficiency (Figure 5a, b): The observed reduction in spectral emissivity within the 11-13 micron range, and the converse for scattering efficiency, warrants a detailed explanation. Could you cite literature that sheds light on these phenomena?

Reflectance Tuning through Particle Distribution: While it may extend beyond the scope of this study, the role of surface roughness in modulating emissivity and reflectance is intriguing. Is there potential for tuning reflectance by manipulating particle distribution or by incorporating a mixture of silver nanoparticles to create effective roughness or flatness?

Cooling Power Discussion: In the discussion on cooling power, it would be valuable to compare the observed performance with current state-of-the-art approaches. Please include relevant literature to contextualize the findings.

Kind regards,

Author Response

(The authors gave the same response as above.)
